# Essential Oils Produce Developmental Toxicity in Zebrafish Embryos and Cause Behavior Changes in Zebrafish Larvae

**DOI:** 10.3390/biomedicines11102821

**Published:** 2023-10-18

**Authors:** Ivanildo Inacio da Silva, Niely Priscila Correia da Silva, James A. Marrs, Pabyton Gonçalves Cadena

**Affiliations:** 1Departamento de Morfologia e Fisiologia Animal (DMFA), Universidade Federal Rural de Pernambuco, Av. Dom Manoel de Medeiros s/n, Dois Irmãos, Recife 52171-900, PE, Brazilni-gold@hotmail.com (N.P.C.d.S.); 2Department of Biology, Indiana University Purdue University Indianapolis, 723 West Michigan, Indianapolis, IN 46202, USA

**Keywords:** toxicology, essential oils, zebrafish, embryogenesis

## Abstract

Essential oils have gained significant popularity in various industries due to their biological properties, but their potential toxic effects on living organisms have been poorly investigated. This study aimed to evaluate the effects of lemongrass, thyme, and oregano essential oils on zebrafish embryos and larvae as animal models. Embryos were exposed to different concentrations of essential oils, and various endpoints were assessed, including epiboly, mortality (LC50), morphometry, and behavioral changes. All three essential oils reduced epiboly, affecting embryonic development. LC50 values were calculated for lemongrass (3.7 µg/mL), thyme (14.4 µg/mL), and oregano (5.3 µg/mL) oils. Larvae exposed to these oils displayed morphological defects, including growth reduction, spinal deformation, pericardial edema, eye size reduction, and reduced swim-bladder inflation. Morphometric analysis confirmed reduced larval length at higher oil concentrations. Essential-oil exposure altered zebrafish larval swimming behavior, with lemongrass oil reducing dark-cycle activity and oregano oil increasing light-cycle activity, suggesting neurodevelopmental toxicity. These findings illustrate the adverse effects of these oils on zebrafish embryos and larvae and reveal essential-oil toxicity, indicating careful use should be considered, particularly during pregnancy.

## 1. Introduction

Essential oils are widely used in the food, cosmetic, and pharmaceutical industries due to their biological properties, such as antimicrobial and antioxidant activities [1]. The widespread use of essential oils and other plant derivatives in phytotherapy, as well as herbal teas and extracts, can have potential harmful effects on human health if not used properly. However, the potential toxic effects of these oils on living organisms have been poorly investigated. The truth is that any medicinal herb can have toxic effects depending on dose and exposure time [2]. Furthermore, the lack of regulation and standardization in the production and marketing of these products makes it difficult to determine the appropriate doses and concentrations for their safe use. In addition, the interaction of these compounds with other drugs and their effects on vulnerable populations, such as pregnant women and children, require further investigation. Therefore, it is important to exercise caution when using plant derivatives for therapeutic purposes and to seek guidance from a qualified healthcare professional to ensure safe and effective treatment.

Among the essential oils, lemongrass (*Cymbopogon flexuosus*), thyme (*Thymus vulgaris*), and oregano (*Origanum vulgare*) oils were reported to exhibit several biological activities, such as anticancer, anti-inflammatory, and antimicrobial effects [3]. These oils are widely used in the food industry as natural preservatives and flavorings, as well as in traditional medicine for various purposes, including digestive disorders, respiratory infections, and wound healing [4]. Despite their potential benefits, essential oils can also have toxic effects at certain concentrations, which can pose a risk to human and environmental health.

The zebrafish (*Danio rerio*) is a well-established model organism for developmental biology and toxicology studies. This model emerges as an attractive, efficient, and economical alternative to mammalian models [5]. Zebrafish have a short life cycle, high fecundity, low maintenance costs, and easy handling, allowing for greater standardization of experiments. Therefore, the zebrafish model fits within the 3Rs principle (reduction, refinement, and replacement), as it reduces the number of animals used in experiments, refines experimental conditions, and replaces the need for the use of large animals, contributing to the reduction of animal suffering and more ethical and sustainable research [6]. The zebrafish external reproduction facilitates embryonic developmental studies, including the toxicological approach, and the early stages of embryonic development are surprisingly similar to human development [7]. Moreover, these fish demonstrate a series of complex behaviors that include social, anxiety, learning and memory, and defensiveness that may be useful in various research studies. Locomotor activity and stimuli response patterns of zebrafish larvae in light and dark conditions are a great resource for testing potential neurotoxic compounds [8]. 

It is remarkable that the zebrafish model has several features that are ideal for toxicology testing with herbal products that have potential pharmacotherapeutic application. It is essential to evaluate the potential toxic effects of these oils and their compounds, particularly during early developmental stages when the organism is more vulnerable. In this study, we investigated the effects of lemongrass, thyme, and oregano oils on the early development of zebrafish embryos and larvae using a comprehensive set of endpoints, including epiboly, embryonic development, mortality, and behavior.

## 2. Materials and Methods

### 2.1. Reagents and Preparation of Stock Solutions

Lemongrass (GT Brazil) (lot#LZ3218), thyme (QT Thymol/GT Hungary) (lot#LZ3010), and oregano (QT Carvacrol/GT Turkey) (lot#LZ3152) essential oils used in the present study were purchased from Laszlo-Aromaterapia Ltda (Belo Horizonte, MG, Brazil). Dimethylsulfoxide (DMSO) (lot#SHBL2891) was purchased from Sigma (St. Louis MO, USA). All essential oils were previously characterized by gas chromatography according to the supplier (Table 1). All other reagents utilized were of analytical grade. Stock solutions of essential oils used were made fresh before the experiments and dissolved in 0.1% (*v*/*v*) of DMSO in an embryo medium [9].

### 2.2. Zebrafish Husbandry, Embryo Collection, and Essential-Oil Exposure

All experiments were performed at the Purdue School of Science at Indiana University–Purdue University Indianapolis (IUPUI). The methods and procedures used in this work are in accordance with the guidelines and supervision of the Indiana University Policy on Animal Care and Use. Adult zebrafish AB strains were raised and maintained under standard laboratory conditions according to OECD 236 guidelines [10]. Adults (2:1 male/female) were separated by a divider in a breeding tank. Early morning on the breeding day, the divider was removed. Fertilized eggs were checked and collected 20 min after crossing and incubated in embryo medium. Only embryos that had normal development were used in the experimental groups. At 2 hpf (hours postfertilization), embryos were transferred to Petri dishes and exposed to lemongrass (1.5, 2.5, 3.5, 4.5, and 6.0 µg/mL), thyme (5.0, 10.0, 15.0, 20.0, and 25.0 µg/mL) and oregano (3.0, 4.0, 5.0, 6.0, and 8.0 µg/mL) essential oils dissolved in DMSO solution with embryo medium (working solutions) and a control group (0.1% DMSO). The concentrations used in this work were determined based on the LC50 values reported in other studies [1,11] and adapted according to the observed response in pilot experiments. Fresh working solutions were made daily for each experimental group. The Petri dishes were washed three times with embryo medium, and working solutions were replaced daily until the end of the experiments. Fish were exposed to working solutions with essential oils until 96 hpf (94 h total). The experimental design is schematized in Figure 1.

### 2.3. Measurement of Epiboly

Epiboly is the spreading of cells over the yolk cell. The percent epiboly measurements were made at 8 hpf of embryos, after 6 h of exposure. Embryos (n = 15 per group with 15 experimental groups = 225 embryos) were fixed overnight in 4% paraformaldehyde (PFA) in phosphate-buffered saline (PBS) solution [12,13]. Images of fixed embryos were captured using a stereomicroscope (Leica DFC 450C) and smartphone camera (MIUI Xiaomi Redmi Note 9) to determine the percent epiboly. The measurements were made according to Cadena et al. [12] using ImageJ Software (version 1.53k, 2019, National Institutes of Health, Bethesda, MD, USA).

### 2.4. Analysis of Embryonic Development

Embryos (n = 40 per group with 15 experimental groups = 600 embryos) were maintained in essential-oil working solutions and analyzed under a stereomicroscope until 6 dpf. Mortality was analyzed under a stereomicroscope daily and confirmed according to OECD 236 [10] in each experimental group. The control-group survival was >90% at the end of all experiments, validating our results [10]. Teratogenic effects were analyzed qualitatively by the capture of images under a stereomicroscope, as described previously (Section 2.3). Observed teratogenic effects were ocular defects; pericardial and multiple edemas in the body; morphological defects on the tail or spine; and defects on the locomotor system [12,14,15,16].

### 2.5. Morphometry

The morphometric evaluation was made using 6 dpf zebrafish larvae (n = 20 per experimental with 9 experimental groups = 180 larvae) fixed in 4% PFA. Only experimental groups with the lowest concentrations for each essential oil were considered due to the high mortality rates at higher concentrations. Lateral images were captured by stereomicroscope as described above. These images were analyzed in ImageJ Software (version 1.53k). Eye length (EL), head length (HL), head width (HW), and fish standard length (SL) were measured as described by Sales Cadena et al. [15].

### 2.6. LC50 Determination

The median lethal concentration (LC50) was obtained using the probit method [17]. LC50 values were calculated using mortality data to produce a linear regression, which preferably has concentrations with 0% and 100% of the response effect (mortality). Therefore, LC50 was evaluated from the survival percentage of zebrafish larvae (6 dpf) exposed to the different concentrations of essential oils studied.

### 2.7. Animal Behavior

The light–dark excitatory locomotor test was performed on 6 dpf larvae [14]. Briefly, dead larvae or those with debilitating morphophysiological damage were not included in this test. Therefore, the experimental groups with more individuals capable of behavior (at least ≈ 24 larvae) were selected. Larvae were placed in 96-square-well plates (with one larva per well) filled with embryo medium. A total of 24 larvae (9 experimental groups = 216 larvae) were used per group (including the control group). ZebraBox (Viewpoint Life Sciences, Lyon, France) was used to track and record. First, all animals were subjected to an acclimation period of 20 min. After this, six light–dark cycles started, with each cycle consisting of 10 min in light (100% light intensity in ZebraBox), followed by 10 min in dark. All larvae locomotion data were tracked and collected. Then, these data were grouped into 2 min periods (integration period) during each light and dark 10 min interval [14]. Distance moved was used as the endpoint, and measurement included two velocities: small (between 4 and 8 mm/s) and large (more than 8 mm/s) activity movement. 

### 2.8. Statistical Analysis

The data are represented by mean ± SD (standard deviation). Differences between the control and essential-oil-exposed groups were tested by one-way ANOVA, followed by Tukey’s test. In the behavior test, each well/fish was considered a repetition. To evaluate the locomotor distance between the groups, when the group and time variables were considered, we used two-way ANOVA, followed by Tukey’s test. The difference was considered significant when *p* < 0.05. The data results were analyzed by GraphPad Prism software (version 8.0.1, 2018, La Jolla, CA, USA).

## 3. Results

### 3.1. Epiboly Measurement

The epiboly (spreading of embryonic cells over the yolk cell) of the embryos was measured at 8 hpf. As expected, DMSO solution (0.1%) exposure did not alter the epiboly compared with embryo medium alone (≈ 75.0% and 75.5%, respectively). As shown in Figure 2, all oils reduced the percent epiboly of embryos at 8 hpf (Figure 2A). The lemongrass oil reduced epiboly at the high concentrations of 3.5, 4.5, and 6.0 µg/mL; however, it did not alter epiboly at 1.5 and 2.5 µg/mL (Figure 2B). The thyme oil exposure decreased the percent epiboly at the concentrations of 10.0, 15.0, 20.0, and 25.0 µg/mL. Only the low concentration (5.0 µg/mL) did not affect the epiboly. Similarly, the 4.0, 5.0, 6.0, and 8.0 µg/mL oregano oil (Figure 2C) reduced the percent epiboly, while the 3.0 µg/mL concentration did not affect epiboly when compared with the control group.

### 3.2. Embryonic Development and Mortality Analysis

The typical morphological defects caused by essential-oil exposure are shown in Figure 3. This figure shows the epiboly reduction when animals were exposed to thyme essential oil in the 25 µg/mL group (Figure 3B) compared with the control group (Figure 3A). The observed teratogenic effects are described in Table 2, indicating their presence or absence in the different experimental groups.

The morphometric analysis of 6 dpf zebrafish larvae shows that all three evaluated oils caused a reduction in the standard length of the larvae at the highest concentrations evaluated (Table 3). In addition, the lemongrass oil group at 3.5 µg/mL reduced the tail width, while both thyme oil (15.0 µg/mL) and oregano oil (5.0 µg/mL) reduced the eye length of the larvae.

During all experiments, less than 10% of mortality was observed in control groups. However, the essential-oil exposure caused a rise in mortality proportion in most of the studied concentrations (Figure 4), except by the lemongrass 1.5 µg/mL, thyme 5.0 µg/mL, and oregano 3.0 µg/mL groups. The survival curve of each oil studied was produced (Figure 4A–C) and was used to calculate the LC50 (at 6 dpf) by the probit method. Therefore, the LC50 calculated was 3.7 µg/mL in lemongrass oil (Figure 4A); 14.4 µg/mL in thyme oil (Figure 4B); and 5.3 µg/mL in oregano oil (Figure 4C).

### 3.3. Behavior

The standard swimming behavior of the fish in the light–dark excitatory locomotor test is shown in Figure 5. Only animals without apparent morphological defects were used in this test. As expected, the control group exhibited more activity during the dark cycle than the light cycle. This behavior pattern was affected by exposure to essential oils. When analyzed separately, the distance traveled in each cycle was reduced during the dark cycle for both small and large activities in larvae exposed to 1.5 and 2.5 µg/mL of lemongrass oil (Figure 5A,B), also reducing the distance of large movements during the light cycle with the concentration of 2.5 µg/mL (Figure 5B). In contrast, embryos exposed to 6.0 µg/mL of oregano oil showed an increase in both small and large activities in both cycles (Figure 5K,L). Interestingly, when the data are plotted as a function of time (2 min) in complete cycles (light–dark = total of 20 min), all oils affected the zebrafish larvae’s swimming behavior pattern. Larvae incubated with lemongrass oil showed fewer small and large movements compared with the control (Figure 5C,D), and this effect was dose-dependent on the average distance traveled in large movements (Figure 5D). The 5.0 µg/mL thyme oil group reduced the average of short movements (Figure 5G), while the 10.0 µg/mL thyme oil group restored this effect at the same level as the control group (Figure 5G). Oregano oil also increased the distance traveled in large movements in a dose-dependent manner (Figure 5L), although a significant effect was only observed in the 6.0 µg/mL group in large movement activity.

## 4. Discussion

Essential oils have been used for centuries in various applications, such as aromatherapy, natural medicine, and perfumery [18]. These oils are derived from different parts of plants and are known for their diverse chemical compositions. It is well established that environmental conditions have a profound influence on the biochemical pathways within plants, leading to variations in essential-oil compositions and their biological effects [19]. 

The gas chromatography of the studied oil demonstrates that the major component of each oil is in accordance with previous data. Lemongrass oil has citral (neral and geranial) as the main constituent, while thymol and carvacrol are the main molecules in thyme oil, and thymol is the major component in oregano oil [20,21,22]. These molecules can pass through zebrafish chorion [23] and probably cause the toxicity observed in our study. The first evidence that these molecules pass through the chorion was the reduction in the percent epiboly found in our study. The reduced percent epiboly is indicative of a delay in the gastrulation process, in this case, probably caused by citral and thymol action. Citral is commonly used in pharmacology as an inhibitor of retinoic acid biosynthesis [24]. This acid plays a fundamental role in regulating early embryogenesis, especially during gastrulation and development of axis and limb formation [25]. Therefore, citral causes retinoic acid deficiency and induces cytoskeletal malformation and developmental delays [26]. Because of this, we have concerns about the use of this oil by pregnant women. On the other hand, thymol is known to downregulate phosphatidylinositol-3 kinase (PI3K) [27]. This protein is necessary for recruiting and activating F-actin structures in embryo epiboly progression. Therefore, reductions in PI3K signaling could lead to alterations in actin dynamics that can disrupt the cytoskeleton organization in epiboly [28].

Although essential oils are considered safe for human consumption, inappropriate use of these compounds can lead to intoxication, especially in children [29]. Our study detected significant morphological damage in zebrafish embryos exposed to commercial essential oils. Previous studies have found similar results with other oils. Piasecki et al. [23] observed morphological abnormalities and developmental impairment in zebrafish embryos exposed to different essential oils obtained from plants of the genus *Cymbopogon*. Oils from these species have citronellol, geraniol, and citral as major components. Another study [30] also observed malformations, such as yolk sac edema, tail and axis malformations, and pericardial edema in zebrafish exposed to thymus oil, supporting our findings. Taken together, essential oils can induce developmental toxicity and mortality in zebrafish embryos. These data highlight the importance of studying the safety and toxicity of these compounds in other models to determine safe levels for human consumption.

This is the first report to study the effect of these essential oils on the behavior of zebrafish larvae. Therefore, one must be careful in how to relate the observed effects to studies with adult animals. A study by Hacke et al. [31] evaluated the anxiolytic properties of *Cymbopogon citratus* extract, essential oil, and its constituents in zebrafish and found that the essential oil and its constituents, particularly geraniol and citral, exhibited significant anxiolytic activity in adult zebrafish. These results may partially explain the reduction in swimming activity observed in this study on zebrafish larvae exposed to lemongrass oil. On the other hand, in a study conducted by Vieira et al. [32], anxiety-like behaviors were observed in zebrafish embryos exposed to thymol, which is consistent with our findings. We found that zebrafish larvae exhibited an increased distance moved, particularly during dark periods. Additionally, a study related to Capatina et al. [33] investigated the neuroprotective effects of oregano essential oil and found that exposure to the essential oil improved the cognitive function of zebrafish with cognitive impairment induced by scopolamine. This protective effect was attributed to the thymus, its modulation of the cholinergic system, and reduced oxidative stress in the brain. These results corroborate the effect of increased motility observed in our study since oregano oil can cause neurostimulation in zebrafish larvae exposed to this oil. Although the comparison is limited, it is possible to conclude that only lemongrass oil had a relevant neurotoxic effect, even comparable with classical models of toxicity, such as fetal alcohol syndrome [13,14].

Despite these promising results, it is important to note that the effects of essential oils on the nervous system are complex and may be influenced by several factors, such as dose, route of administration, and individual variability. Additionally, it is important to note that although it cannot be directly extrapolated to human health risk assessments, the misuse and overuse of these oils could be harmful. It is crucial to follow proper dilution guidelines, avoid ingesting essential oils, and be aware of potential drug interactions and toxicities, especially when used during pregnancy and childhood.

## 5. Conclusions

In conclusion, the present study provides evidence that exposure to lemongrass, thyme, and oregano oils can induce notable developmental toxicity and mortality in zebrafish embryos. Moreover, these oils were observed to elicit diverse behavioral effects, which may indicate a neurodevelopmental impact. These findings underscore the need for further research to establish safety thresholds and assess potential human health risks associated with improper use of essential oils. Additionally, it highlights the significance of assessing the toxicity and potential disruptive effects of these compounds on aquatic ecosystems.

## Figures and Tables

**Figure 1 biomedicines-11-02821-f001:**
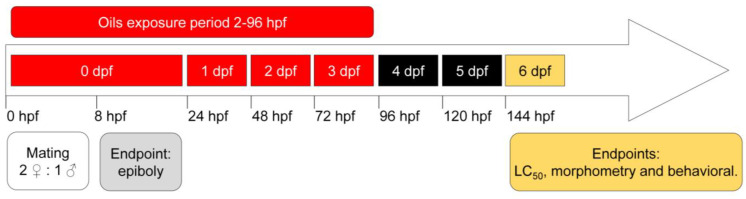
Schematic representation of tests performed for toxicity evaluation of plant essential oils in embryo and larvae zebrafish.

**Figure 2 biomedicines-11-02821-f002:**
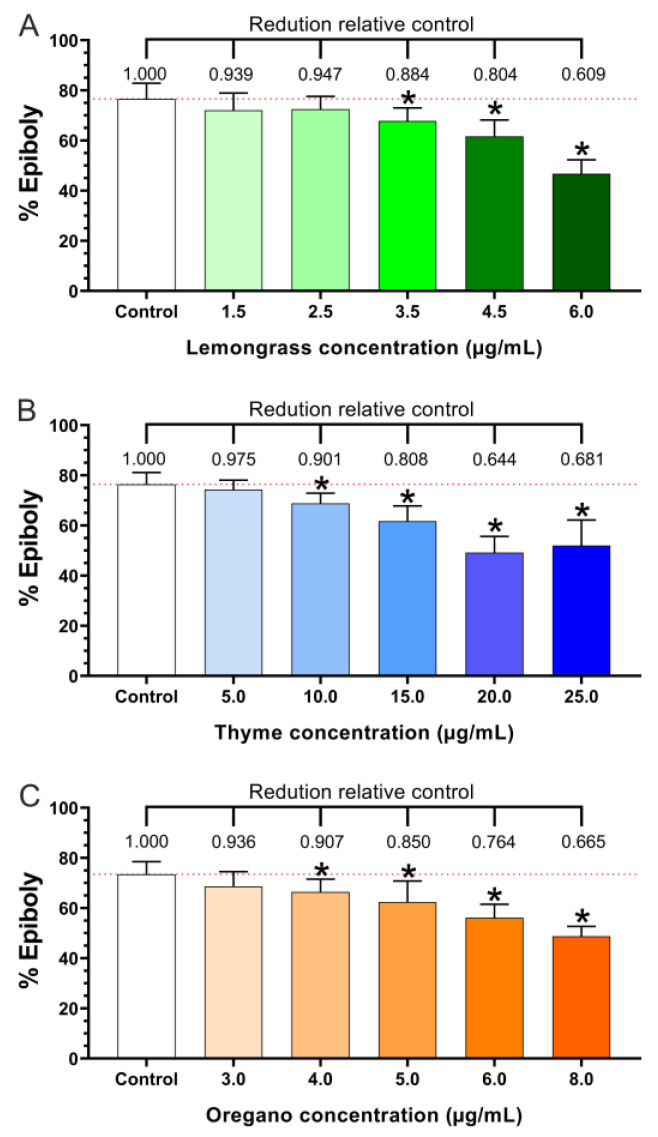
Epiboly percentages at 8 hpf in zebrafish embryos exposed to (**A**) lemongrass, (**B**) thyme, and (**C**) oregano essential oils. Test F by one-way ANOVA: lemongrass EO (F(5, 77) = 48.83, *p* < 0.05); thyme EO (F(5, 76) = 48.97, *p* < 0.05); and oregano EO (F(5, 72) = 29.95, *p* < 0.05). * Statistically different from the control group (one-way ANOVA test, *p* < 0.05).

**Figure 3 biomedicines-11-02821-f003:**
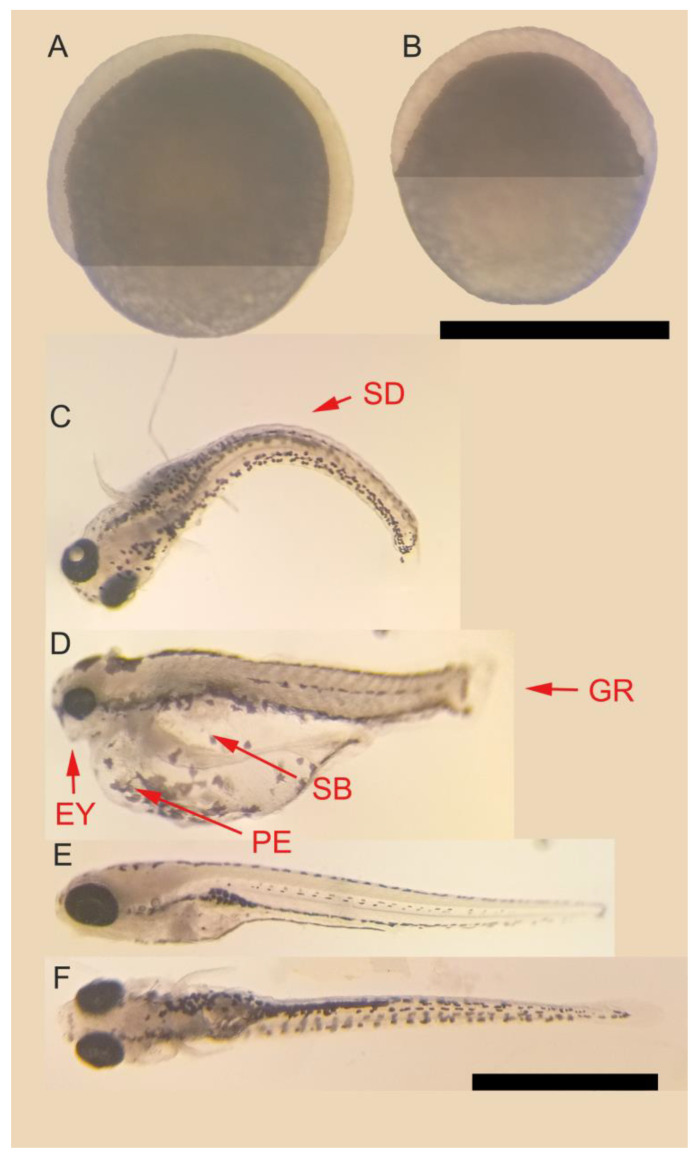
Representative zebrafish embryo epiboly and typical teratogenic effects observed in zebrafish embryos exposed to lemongrass, thyme, and oregano essential oils. Legend: (**A**) Control group epiboly at 8 hpf. (**B**) Thyme 25 µg/mL group epiboly at 8 hpf (scale bar 500 μm). (**C**,**D**) Fish affected by thyme essential-oil (20 µg/mL) exposure. Arrows and abbreviations indicate the main teratogenic effects observed as spine deformation (SD), small eyes (EY), pericardial edema (PE), swim-bladder inflation (SB), and growth retardation (GR). (**E**,**F**) Control group in lateral and dorsal views, respectively (scale bar 1 mm).

**Figure 4 biomedicines-11-02821-f004:**
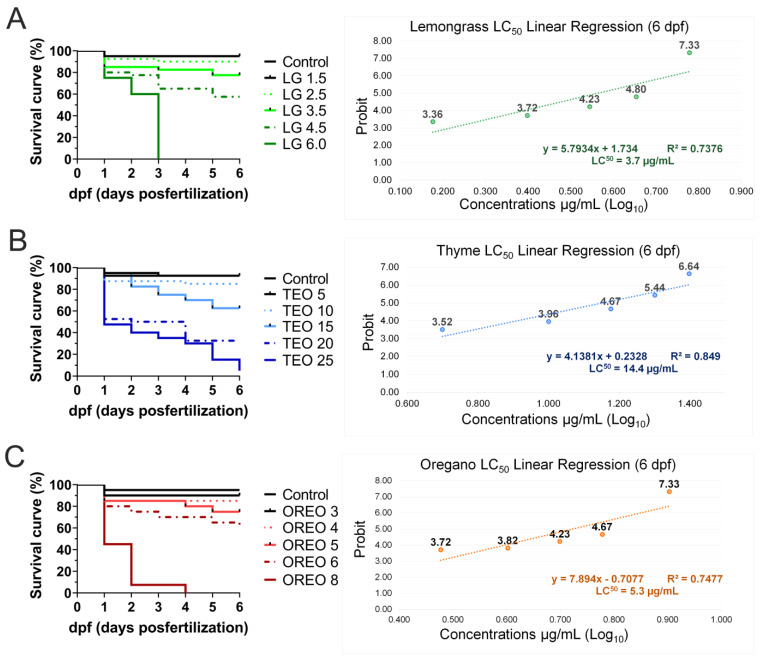
Survival curve and the mortality linear regression of zebrafish embryos after exposure to lemongrass (LGEO), thyme (TEO), and oregano (OREO) essential oils (**A**, **B**, and **C**, respectively). The linear regression and relationship of probit essential-oil concentrations were used to calculate LC50 values in larvae at 6 days post-fertilization.

**Figure 5 biomedicines-11-02821-f005:**
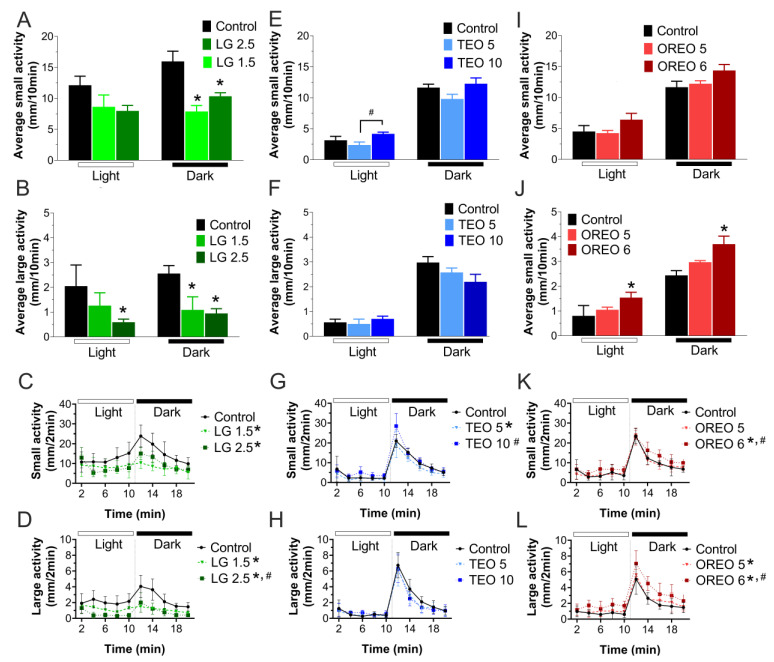
Zebrafish larvae behavior in excitatory dark–light locomotor test. (**A**–**F**) Bars show the average of small (between 4 and 8 mm/s) or large (>8 mm/s) activity movement for each dark or light period (10 min). (**G**–**L**) Average swimming behavior (2 min) in each complete dark–light cycle (20 min). Behavior test was analyzed by one-way ANOVA in light period (lemongrass EO small activity (F(2, 15) = 2.273, *p* = 0.14), large activity (F(2, 15) = 3.943, *p* < 0.05); thyme EO small activity (F(2, 15) = 3.463, *p* = 0.06), large activity (F(2, 15) = 0.5043, *p* = 0.61); oregano EO small activity (F(2, 15) = 1.984, *p* = 0.17), large activity (F(2, 15) = 4.964, *p* < 0.05)) and dark period (lemongrass EO small activity (F(2, 15) = 13.10, *p* < 0.05), large activity (F(2, 15) = 12.77, *p* < 0.05); thyme EO small activity (F(2, 15) = 2.829, *p* = 0.09), large activity (F(2, 15) = 2.547, *p* = 0.11); oregano EO small activity (F(2, 15) = 2.925, *p* = 0.08), large activity (F(2, 15) = 8.370, *p* < 0.05)). When the time variable was considered, two-way ANOVA test was realized in small activity: lemongrass EO small activity (F(2, 69) = 9.015, *p* < 0.05) and large activity (F(2, 69) = 9.924, *p* < 0.05); thyme EO small activity (F(2, 69) = 1.436, *p* = 0.24) and large activity (F(2, 69) = 0.456, *p* = 0.64); oregano EO small activity (F(2, 69) = 1.951, *p* = 0.15) and large activity (F(2, 69) = 2.746, *p* = 0.07). * Represents statistically significant (*p* < 0.05) difference by control group. ^#^ Represents statistically significant (*p* < 0.05) difference by another essential-oil group.

**Table 1 biomedicines-11-02821-t001:** Chemical compositions of commercial lemongrass (*Cymbopogon flexuosus*), thyme (*Thymus vulgaris*), and oregano (*Origanum vulgare*) essential oils. Data (gas chromatography) were provided by the supplier.

Oil	Compound	Amount (%)
Lemongrass (*Cymbopogon flexuosus*)	geranial	46.0
	neral	28.6
	geraniol	6.8
	citronellol	3.7
	linalyl acetate	1.8
	α-terpineol	1.8
	1.8-cineole	1.6
	linalool	1.6
	others	8.1
Thyme (*Thymus vulgaris*)	thymol	29.5
	carvacrol	28.2
	geraniol	9.0
	p-cymene	5.6
	β-caryophyllene	4.3
	γ-terpinene	3.4
	1.8-cineole	3.2
	α-humulene	2.1
	others	14.7
Oregano (*Origanum vulgare*)	thymol	39.7
	β-bisabolene	8.0
	p-cymene	7.6
	γ-terpinene	6.4
	thymol methyl ether	5.0
	bicyclogermacrene	4.0
	δ-cadinene	3.5
	aromadendrene	3.4
	others	22.4

**Table 2 biomedicines-11-02821-t002:** Teratogenic effects observed in 6 dpf zebrafish embryos exposed to lemongrass, thyme, and oregano essential oils.

Oil	Teratogenic Effect	Oil Concentration (µg/mL)				
		1.5	2.5	3.5	4.5	6.0
Lemongrass (*Cymbopogon flexuosus*)	Spine deformation (SD)	-	+	+	+	*
	Small eyes (EY)	-	-	-	+	*
	Pericardial edema (PE)	-	-	-	+	*
	Swim-bladder inflation (SB)	-	-	-	+	*
	Growth retardation (GR)	-	-	-	+	*
		5.0	10.0	15.0	20.0	25.0
Thyme (*Thymus vulgaris*)	Spine deformation (SD)	-	-	+	+	+
	Small eyes (EY)	-	-	+	+	+
	Pericardial edema (PE)	-	-	-	+	+
	Swim-bladder inflation (SB)	-	-	-	+	+
	Growth retardation (GR)	-	-	-	+	+
		3.0	4.0	5.0	6.0	8.0
Oregano (*Origanum vulgare*)	Spine deformation (SD)	-	-	+	+	*
	Small eyes (EY)	-	-	+	+	*
	Pericardial edema (PE)	-	-	-	-	*
	Swim-bladder inflation (SB)	-	-	-	-	*
	Growth retardation (GR)	-	-	-	+	*

+ Indicates presence; - Indicates absence; * All embryos were dead.

**Table 3 biomedicines-11-02821-t003:** Morphometric measurements of zebrafish larvae exposed to different concentrations of lemongrass, thyme, and oregano essential oils. Data are represented by the relative percentage of the control group. Test F by one-way ANOVA in lemongrass EO (eye length (F(19, 57) = 0.5558, *p* = 0.92); rump length (F(19, 57) = 0.9023, *p* = 0.58); rump anus width (F(19, 57) = 1.015, *p* = 0.46); standard length (F(19, 57) = 1.426, *p* = 0.15); tail width (F(19, 57) = 1.228, *p* = 0.27)), thyme EO (eye length (F(19, 57) = 1.096, *p* = 0.38); rump length (F(19, 57) = 0.9803, *p* = 0.50); rump anus width (F(19, 57) = 1.102, *p* = 0.37); standard length (F(19, 57) = 1.053, *p* = 0.42); tail width (F(19, 57) = 0.2940, *p* = 0.99)), and oregano EO (eye length (F(19, 57) = 0.5429, *p* = 0.93); rump length (F(19, 57) = 1.448, *p* = 0.14); rump anus width (F(19, 57) = 1.079, *p* = 0.40); standard length (F(19, 57) = 2.142, *p* = 0.01); tail width (F(19, 57) = 1.449, *p* = 0.14)).

Oil	Measurement	Experimental Group			
		Control	1.5 µg/mL	2.5 µg/mL	3.5 µg/mL
Lemongrass (*Cymbopogon flexuosus*)	Eye length	100.0 ± 6.5	99.4 ± 3.6	100.1 ± 7.2	95.2 ± 7.8
	Rump length	100.0 ± 5.0	101.7 ± 9.2	100.9 ± 3.6	98.9 ± 5.4
	Rump anus width	100.0 ± 3.5	102.6 ± 4.1	98.4 ± 4.3	98.8 ± 5.0
	Standard length	100.0 ± 3.5	98.1 ± 3.7	99.3 ± 4.3	90.6 ± 7.8 *
	Tail width	100.0 ± 7.1	100.1 ± 5.1	98.4 ± 6.3	93.1 ± 5.8 *
Thyme (*Thymus vulgaris*)		Control	5.0 µg/mL	10.0 µg/mL	15.0 µg/mL
	Eye length	100.0 ± 5.4	98.2 ± 4.2	99.0 ± 5.7	92.1 ± 7.3 *
	Rump length	100.0 ± 8.6	99.8 ± 7.7	101.7 ± 4.9	100.7 ± 7.9
	Rump anus width	100.0 ± 3.8	99.4 ± 4.4	100.5 ± 3.4	101.2 ± 6.7
	Standard length	100.0 ± 5.2	99.4 ± 3.7	96.9 ± 5.5	95.0 ± 5.0 *
	Tail width	100.0 ± 4.2	99.6 ± 4.6	96.5 ± 3.7	97.1 ± 5.4
Oregano (*Origanum vulgare*)		Control	3.0 µg/mL	4.0 µg/mL	5.0 µg/mL
	Eye length	100.0 ± 6.6	95.7 ± 4.7	95.6 ± 5.1	93.1 ± 6.9 *
	Rump length	100.0 ± 6.8	103.5 ± 7.7	103.7 ± 5.2	101.8 ± 8.6
	Rump anus width	100.0 ± 3.9	99.5 ± 4.7	100.3 ± 4.2	102.0 ± 3.0
	Standard length	100.0 ± 4.9	99.8 ± 5.0	101.5 ± 6.2	93.1 ± 8.1 *
	Tail width	100.0 ± 6.0	101.5 ± 6.4	97.7 ± 7.5	96.3 ± 5.5

* Statistically different from the control group (Tukey’s HSD test, *p* < 0.05).

## Data Availability

The data and materials used in this study are available from the corresponding author on request.

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
