# Peer review of "Essential Oils Produce Developmental Toxicity in Zebrafish Embryos and Cause Behavior Changes in Zebrafish Larvae"

_biomedicines, 2023, doi:10.3390/biomedicines11102821_

Round 1
Reviewer 1 Report
please, see the uploaded file

Author Response
We are grateful for your careful review and critique of the manuscript. Your concerns are addressed below.
- On which bases did the Authors select the concentrations used in the manuscript? Could the Authors comment if there is a relationship with the oil concentrations used and those present in food or cosmetics and other applications? This should also be discussed in the discussion section.
Initially, the concentrations used in this manuscript were determined based on the LC50 values reported in other studies that evaluated the toxic effects of essential oils in zebrafish (Jayasinghe & Jayawardena, 2019; Thitinarongwate et al. 2021). Therefore, four concentrations (1, 5, 50, and 100 µg/mL) were initially tested, and the final concentrations (reported) were adjusted according to the observed mortality response for each essential oil. This information is added in the paper, lines 98-101.
References:
Jayasinghe, C.D.; Jayawardena, U.A. Toxicity Assessment of Herbal Medicine Using Zebrafish Embryos: A Systematic Review. Evidence-Based Complementary and Alternative Medicine 2019, 2019, 1–17, doi:10.1155/2019/7272808.
Thitinarongwate, W.; Mektrirat, R.; Nimlamool, W.; Khonsung, P.; Pikulkaew, S.; Okonogi, S.; Kunanusorn, P. Phytochemical and safety evaluations of Zingiber ottensii Valeton essential oil in zebrafish embryos and rats. Toxics 2021, 9, 102-119, https://doi.org/10.3390/toxics9050102.
- Line 164-165: “As expected, DMSO (0.1%) exposure did not alter the epiboly of the control group (~75.5%).” If no data comparing DMSO 0.1% and Fish water have been collected, please, cite a specific reference.
The data was collected and analyzed, but not shown on paper. This information was corrected in lines 166-167.
- Line 173-179: the Authors explain that in Fig 3 are reported representative examples of the different morphological abnormalities of zebrafish embryos and larvae exposed to essential oils (and refer to thyme essential oil at 25 µg/ml). Please, check and improve the figure 3 caption, as it is indicated in panel B 20 µg/ml of thyme oil (this differs from the text) and it is not clear which is the concentration and which oils were used in panels C and D.
Thank you, this was corrected. The correct value is 25 µg/mL. Panel C and D are referents to thyme essential oil (20 µg/ml).
- Furthermore, it would be useful for the reader to have a table recapitulating the main teratogenic effects as function of the concentration used.
We add the table 2 according to your suggestions. Thank you.
- Table 2: only the concentrations of lemon grass oil are reported; furthermore, data appear duplicated for all essential oils (please check).
Thank you. All table information was corrected.
- Figure 4: please check the letters identifying panels and change them according to the text.
Thank you. This information was corrected.
- Figure 5: Please check letters identifying panels and change them according to the text. In panel I and J the scales are express in mm/2min, but should be mm/10min, as the graphs refer to the average of the small and large activity for each light and dark period.
Thank you. The letter information was corrected.
- The control fish in the lemon grass tests behave slightly different than the controls in the experiments testing thyme and oregano oils. In particular, the activity measured during the light period is higher. Is there any reason for this behaviour? Panel G: are the Authors sure that the TEO 5 µM in small movements is statistically different from the control? The same in panel L for OREO 5 µM in large movements.
The control group data are correctly represented. The assays with different oils were performed independently (on different days), so it is expected that there will be variations between them. However, these variations do not affect the behavioral analysis, as all factors are controlled within each assay.
On panels G and L, we also checked the data and confirmed that there is a statistical difference between them.
- As a final comment: these behavioural differences may be statistically significant, but are they really biologically significant?
Although the graphic data are not visually different, the statistical analysis indicates that there is a difference in animal behavior (as evidenced in the motion capture videos). For example, animals exposed to LGEO frequently assume a curved posture (as in Fig 3C) and swim in circles, achieving a shorter displacement than the control.
Reviewer 2 Report
biomedicines-2637549 clearly presents its aim of study and experimental design. All figures are nicely presented except that the background color of figure 3 can be better adjusted for readers to have a better view of changes in epiboly and zebrafish body structures. The overall quality of this submission reaches the level of acceptance and it can be accepted for publication after improving the background color of figure 3. The title of this article can be changed to ---produce developmental toxicity in zebrafish embryos and cause behavior changes in zebrafish larvae.
Author Response
We would like to thank you for your careful review and critique of the manuscript. Your concerns are answered below.
- All figures are nicely presented except that the background color of figure 3 can be better adjusted for readers to have a better view of changes in epiboly and zebrafish.
Unfortunately, the yellow microscope light prevents us from changing the background without significantly altering the original images. We have collected all the data under these circumstances.
- The title of this article can be changed to ---produce developmental toxicity in zebrafish embryos and cause behavior changes in zebrafish larvae.
We changed the title according to your suggestions. Thank you.